# Evaluation of the Effectiveness of Telemedicine Visits in a Pediatric Gastroenterology Service in the Context of COVID-19

**DOI:** 10.3390/ijerph192315999

**Published:** 2022-11-30

**Authors:** Cristina Adroher Mas, Celia Calvo Aroca, Ricard Casadevall Llandrich, Francesc López Seguí, Javier Martin Carpi, Francesc García-Cuyàs

**Affiliations:** 1Sant Joan de Déu Hospital, Catalan Ministry of Health, 08950 Barcelona, Spain; 2Center for Research in Health and Economics, Pompeu Fabra University (CRES-UPF), 08002 Barcelona, Spain

**Keywords:** gastroenterology, pediatrics, effectiveness, non-face-to-face care, health economics, telemedicine, COVID

## Abstract

Background: During the first months of COVID-19, the Gastroenterology, Hepatology and Nutrition service of the Hospital Sant Joan de Déu in Barcelona, a leading pediatric center in Spain, introduced a new model of non-face-to-face care. Objective: To evaluate the impact of telephone consultations compared to those conducted face-to-face on healthcare utilization. Methodology: Two main indicators of effectiveness are used: the degree of resolution (percentage of first telemedicine visits that did not generate any new visits in the following 4 and 12 months) and the average number of subsequent visits. A distinction was made between visits for general pathologies (less complex) and those for pathologies treated in monographic consultations (chronic or complex pathologies). Effectiveness at 4 and 12 months was also compared. Results: After 4 months from the first visit, the degree of resolution is lower in the first telemedicine visits than in face-to-face visits for both general pathologies and those of monographic agendas for chronic and complex pathologies. After twelve months, the first general telemedicine visits are less resolute than face-to-face visits, while the resolution rate is the same for chronic and complex pathology visits. Each telemedicine visit generates on average more visits than face-to-face visits. In the short term, 133.4% more in the case of general visits and 51.4% more in the case of chronic and complex visits. In the long term, general telemedicine visits generate 57.31% more visits, while no statistically significant difference is observed between chronic and complex face-to-face and telemedicine visits. Conclusion: The results of this study show that the resolution capacity of the non-face-to-face model in pediatric care in the pandemic context is generally lower and generates more successive visits than the face-to-face model. This lower performance of the telemedicine model should be counterbalanced with its advantages.

## 1. Introduction

Telemedicine is defined as the provision of health care services in which the interaction between patient and health care professional takes place at distance using ICTs [1]. A recent systematic review of randomized controlled trials on the use of telemedicine in pediatrics shows that outcomes are comparable to those of face-to-face visits and, in some cases, may be more effective [2].

Telemedicine can reduce costs (consultation, travel and time) and improve clinical outcomes, and is therefore generally perceived and regarded as cost-effective [1]. A systematic review of the economic evidence of telemedicine shows that it can be cost-effective when applied to different specialties such as pulmonary cardiology or ophthalmology [3,4]. Some studies suggest that reductions in waiting time, travel costs, absence from work and, in the case of pediatric patients, absence from school, are the main drivers of savings from this type of care [5,6]. Furthermore, some articles suggest that telemedicine can be effective in treating diseases in the field of gastroenterology [7]. In economic terms, evidence indicates that telemedicine used to treat conditions such as inflammatory bowel disease is cost-saving and has a high probability of being cost-effective [8]. In addition, patient satisfaction is high for both children and their families [6,9]. In the field of gastroenterology, previous studies suggest that in outpatient gastroenterology referrals, 71% of referrals can be resolved without a face-to-face visit [10].

On 14 March 2020, the Spanish government decreed a lockdown to deal with the COVID-19 pandemic. Healthcare centers had to adapt to these circumstances, in many cases opting to change rapidly, and as far as possible, the modality of the visits they had scheduled, moving them to a telemedicine format. This led to a considerable increase in the use of telemedicine: in Catalonia, and in the context of Primary Care, non-face-to-face visits increased by 267.76% between 2019 and 2020. Although the frequency in visits with diagnoses linked to endocrinology, nutritional aspects and metabolic diseases decreased by 22.27%, they increased by 157.07% in the non-face-to-face setting, being one of the types that increased the most between 2019 and 2020 [11]. During this period, the Hospital Sant Joan de Déu in Barcelona, a pediatric center of reference at the Spanish level, all visits were rescheduled to non-face-to-face format. The starting point was 18% of non-face-to-face visits before the pandemic, achieved as part of the hospital’s digital transformation strategy. In line with this strategy, at the beginning of the lockdown, a pilot test of non-face-to-face care was initiated in the Gastroenterology, Hepatology and Nutrition service, introducing a new model of direct and synchronous care between professional and patient, whereby the first visit of a patient referred from primary care would always be carried out through telemedicine and, thereafter, the professional would decide whether the patient needed a second visit and whether this should be face-to-face or not.

In this context, the aim of this article is to evaluate the effectiveness of the non-face-to-face care model of the Sant Joan de Déu Hospital of Barcelona compared to the conventional face-to-face model, measured in terms of its resolution capacity and the number of successive visits generated.

## 2. Materials and Methods

This study uses administrative data from the first telemedicine visits of the Gastroenterology, Hepatology and Nutrition service performed between 1 June 2020 and 31 March 2021 (964 visits). In these, only those patients who had not previously visited the service were included. Likewise, nursing visits, dietary visits and those related to diagnostic tests were excluded (therefore, only medical visits are considered), as well as patients who, during this period, were visited in person before their first telemedicine visit.

A distinction was made between visits for general pathologies (those with a theoretically rapid resolution and which are expected to be returned to primary care in a few visits) and those for chronic and complex pathologies which are expected to require long-term outpatient follow-up. Those patients who were visited for both types of pathologies (or in pathologies of other types) have been excluded from the study in order to obtain a better comparison of the two groups of patients.

The study reports two sets of analyses, one for the degree of resolution and a second for the average numbers of follow-up visits. The degree of resolution is defined as the percentage of first visits that did not generate subsequent visits. Eight degrees of resolution were calculated recognizing the differences between telemedicine and face-to-face visits, the 4 and 12 months (121 and 365 days, respectively) post-initial visit, and general versus chronic and complex pathologies. The average number of visits per patient was also calculated for each of the same 8 categories.

These results have been compared with those of the first face-to-face visits carried out between 1 June 2018 and 31 March 2019 (1098 first face-to-face visits). As a point of reference, the period two years ago was used, in order to avoid including the first months of the COVID-19 pandemic, months in which the activity of the service decreased. Although the first period was based on the face-to-face visit and the second on the telemedicine model, both periods had both visit options. To test whether the difference between the two periods is statistically significant, a *t*-test was performed for differences in means, a Mann–Whitney test for differences in the population and a Chi-square test for differences in proportions, with significance levels of 95% and 99%.

## 3. Results

Table 1 shows a summary of the patients and their visits for each model (face-to-face and telemedicine), divided between general and chronic and complex types of visits.

Table 2 shows that the degree of resolution in the short term (four months after the first visit) is lower in the first telemedicine visits than in the face-to-face visits, both for general visits (58% vs. 81%) and for chronic and complex pathologies (41% vs. 51%). However, if the degree of long-term resolution (one year after the first visit) is analyzed, it is observed that while the first general telemedicine visits are less resolved than face-to-face visits (24% vs. 39%), in the case of chronic and complex pathologies the resolution is the same in both models of care (23% vs. 24%, not being a statistically significant difference). These results suggest that the telemedicine model is less effective than the face-to-face model in the short term, but that these differences are smaller in the medium term. In this case, the non-face-to-face model of care in chronic and complex pathology visits is just as effective as the conventional model.

Table 3 shows the number of visits generated from the first visit. In one year, general telemedicine first visits generated 73.37% more consultations than face-to-face visits. Chronic and complex pathologies’ visits also generated more consultations when started with a telematic visit, but the increase (30.07%) was lower than for the general visits.

In the short term, both types of telemedicine visits generate more visits than face-to-face visits on average: 133.4% more in the case of general visits and 51.4% more in the case of chronic and complex visits. Between the fifth and the twelfth month after the first visit, general telemedicine visits generate 57.31% more visits than face-to-face visits. On the other hand, in the case of chronic and complex pathologies, there is no statistically significant difference between face-to-face and telemedicine visits. However, this lack of difference does not compensate the difference in the short term, and the overall average of visits generated in the long run (in one year) is higher in the telemedicine model, as stated before. In general, these results are consistent with Table 2: the first telemedicine visits being less resolute, they generate a higher average number of successive visits, and the difference is lower for chronic and complex pathologies than for general ones.

Table 4 shows that the number of successive visits of the patients who were attended with the telemedicine model is different from that of the patients attended with the face-to-face model, both for the ones with general pathologies and the ones with chronic and complex ones. That is also the case for the number of successive visits in the short term. However, between the fifth and the twelfth month, only the number of successive visits for general pathologies is different in the two models, while we the difference for chronic and complex pathologies is not statistically significant. These results are consistent with the ones in Table 3, and overall consistent with the ones in Table 2.

The results also show that 79.48% of new visits generated from a first telemedicine visit are also telemedicine visits (73.83% in the short term; 83.81% in the long term). In the case of general pathologies, 85.96% are performed through telemedicine, a higher percentage than for chronic and complex pathologies (76.74%). First, these results suggest that more complex pathologies are more likely to end up being face-to-face, as well as short-term visits are more likely to be face-to-face than long-term visits. Secondly, they confirm that in the COVID context, non-face-to-face care was the preferred type of care, with face-to-face care reserved for cases that are more complex.

## 4. Discussion

One of the main advantages of telemedicine is the greater accessibility, time savings and lower travel costs for patients (and, especially in pediatric specialties, for their families), as well as the opportunity cost that, in the pediatric context, absence from school implies [6]. In addition, this model of care is particularly useful in facilitating decreased exposure of frail patients to the hospital environment during the COVID-19 pandemic. Evidence also shows that patient satisfaction with non-face-to-face models of care is high, especially among the young population [9,12,13].

Despite these advantages, it is important to consider the effectiveness of this type of care relative to face-to-face care. Generally, studies analyzing the effectiveness of telemedicine use utility or clinical outcome measures, suggesting in which circumstances it may be most useful [2,14]. There are few studies that measure effectiveness in terms of the level of visit resolution [15,16,17], and even fewer that compare face-to-face and telemedicine models of care. The results of this study show that the resolution capacity of the non-face-to-face model is generally lower and generates more successive visits than the face-to-face model. This lower performance of the telemedicine model should be counterbalanced with its advantages.

Contrary to the authors’ initial hypothesis, the telemedicine model is more effective when dealing with chronic and complex pathologies than with general ones. Nearly 25% of patients with chronic and complex pathologies do not need a second visit within a year from the first one, in both the face-to-face and telemedicine models. Nonetheless, the first visits in the telemedicine model generate 30.07% more visits during the first year after the first visit.

This study has several limitations. On the one hand, the care model used for comparison may not be identical in terms of its idiosyncrasies: although the sample compared is similar in number of observations, the pandemic care setting in which the performance of the telemedicine model is observed is certainly different from the normal care setting.

The application of the non-face-to-face model in the pandemic context of COVID-19 has allowed the Hospital Sant Joan de Déu to continue to care for patients in the Gastroenterology, Hepatology and Nutrition service. It is possible that pediatric specificity poses a greater challenge when implementing a telemedicine model, since the interlocution with the patient may be more complicated, making it more difficult to gather information on the patient’s condition in a first telemedicine visit. In this regard, the training of professionals should be ensured in order to maximize the effectiveness of this model of care and to define a strategy for the deployment of the non-face-to-face care model. Because it occurred in the pandemic context, the non-face-to-face model was adopted suddenly. It is worth considering that adopting the model with prior preparation could improve its effectiveness. Finally, in the present study, telemedicine visits were mostly by telephone. Future expansion of the use of videoconferencing could improve its effectiveness as well.

## 5. Conclusions

The results of this study show that in a COVID-19 context, the resolution capacity of the non-face-to-face model of care in a pediatric Gastroenterology, Hepatology and Nutrition Service is generally lower and generates more successive visits than the face-to-face model. This lower performance of the telemedicine model should be counterbalanced by its advantages: greater accessibility, time savings and lower travel costs, reduced exposure to the hospital environment in the COVID context, and greater satisfaction for patients and families.

## Figures and Tables

**Table 1 ijerph-19-15999-t001:** Summary of the patients in the face-to-face and telemedicine models and their visits.

	Face-to-Face Model		Telemedicine Model	
	General	Chronic and Complex	General	Chronic and Complex
Number of patients	294	804	358	606
Total number of visits	564	2181	928	1956
Average successive visits per patient	0.92	1.71	1.59	2.23
Median of successive visits per patient (IQR)	1 (0–2)	1 (1–2)	2 (1–2)	2 (1–3)
Number of patients with no successive visits (%)	116 (39.46%)	196 (24.38%)	87 (24.30%)	139 (22.94%)
Number of patients with 1 successive visit (%)	102 (34.69%)	264 (32.84%)	84 (23.46%)	147 (24.26%)
Number of patients with 2 successive visits (%)	63 (21.43%)	174 (21.64%)	103 (28.77%)	133 (21.95%)
Number of patients with 3 successive visits (%)	11 (3.74%)	87 (10.82%)	62 (17.32%)	75 (12.38%)
Number of patients with 4 successive visits (%)	1 (0.34%)	32 (3.98%)	16 (4.47%)	52 (8.58%)
Number of patients with 5 or more successive visits (%)	1 (0.34%)	51 (6.34%)	6 (1.68%)	60 (9.90%)

**Table 2 ijerph-19-15999-t002:** First visits’ degree of resolution.

	Face-to-Face Model	Telemedicine Model	Difference in %
In the first 4 months after the first visit			
General pathologies	239/294 (81.29%)	206/358 (57.54%)	−23.75% **
Chronic and complex pathologies	411/804 (51.12%)	250/606 (41.25%)	−9.87% **
In a year after the first visit			
General pathologies	116/294 (39.46%)	87/358 (24.3%)	−15.15% **
Chronic and complex pathologies	196/804 (24.37%)	139/606 (22.94%)	−1.44%

* Significant at a 95% level. ** Significant at a 99% level.

**Table 3 ijerph-19-15999-t003:** Mean of visits generated by each first visit.

	Face-to-Face Model	Telemedicine Model	Mean Difference (99% CI)	% Variation
Successive visits (Total successive visits in the first year after the first visit)				
General pathologies	0.92	1.59	0.67 ** (0.45, 0.9)	73.37%
Chronic and complex pathologies	1.71	2.23	0.52 ** (0.18, 0.85)	30.07%
Short-term visits (in the first 4 months after the first visit)				
General pathologies	0.19	0.45	0.26 ** (0.16, 0.36)	133.4%
Chronic and complex pathologies	0.73	1.11	0.38 ** (0.17, 0.58)	51.4%
Successive visits between the fifth and the twelfth month after the first visit				
General pathologies	0.72	1.14	0.42 ** (0.22, 0.61)	57.31%
Chronic and complex pathologies	0.98	1.12	0.14 (−0.05, 0.33)	14.18%

* Significant at a 95% level. ** Significant at a 99% level.

**Table 4 ijerph-19-15999-t004:** Median of visits generated by each first visit.

	Face-to-Face Model	Telemedicine Model	*p*-Value
	Median (IQR)	Median (IQR)	
Successive visits (Total)			
General pathologies	1 (0−2)	2 (1−2)	1.348 × 10^−12^ **
Chronic and complex pathologies	1 (1−2)	2 (1−3)	0.0002636 **
Short-term visits (in the first 4 months)			
General pathologies	0 (0−0)	0 (0−1)	7.274 × 10^−11^ **
Chronic and complex pathologies	0 (0−1)	1 (0−1)	6.975 × 10^−6^ **
Long-term visits (between the fifth and the twelfth month)			
General pathologies	1 (0−1)	1 (0−2)	1.33 × 10^−6^ **
Chronic and complex pathologies	1 (0−1)	1 (0−2)	0.2834

* Significant at a 95% level. ** Significant at a 99% level.

## Data Availability

Not applicable.

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
