# Peer review of "Evaluation of the Effectiveness of Telemedicine Visits in a Pediatric Gastroenterology Service in the Context of COVID-19"

_ijerph, 2022, doi:10.3390/ijerph192315999_

Round 1

Reviewer 1 Report (New Reviewer)

Telemedicine is important style in modern medical practice, especially under the Covid-19 crisis. As mentioned by the authors, the telemedicine visits were mostly carrid out by phone in the study. I suggest that telemedicine visits through video may be more effective and more worthy to be studied. The writing also need some intentive checking and improving. Despite these, the manuscript can be considered to publish.

The following are points that may need improvement (including but not limited to):

1. In the section 2, "general pathologies" and chronic and comples pathologies" were not clearly stated and the role of these in the study and in patients' care.

2. 4 and 12 months after the first visit were chosen as check points, the rationale of this may need to be better clarified in introduction or in discussion section.

3. "resolution" may need to be defined clearly?

4. The title of Talbe 1 seemed misising.

5. What is the difference between tab 3 and tab 4 with totally same title?

6. The discussion may need to be strengthened, especially the specific manner and their advantages and disadvantages of telemedicine?

Author Response

Dear reviewer, thank you for taking the time to review our study. We are going to answer your points one by one:

Telemedicine is important style in modern medical practice, especially under the Covid-19 crisis. As mentioned by the authors, the telemedicine visits were mostly carrid out by phone in the study. I suggest that telemedicine visits through video may be more effective and more worthy to be studied. The writing also need some intentive checking and improving. Despite these, the manuscript can be considered to publish.

The following are points that may need improvement (including but not limited to):

  1. In the section 2, "general pathologies" and chronic and comples pathologies" were not clearly stated and the role of these in the study and in patients' care.

The authors have rewritten the paragraph related to this topic, and have added examples of each kind of pathology, as well as they have justified why they have chosen to make a distinction between both types of visits. 

  1. 4 and 12 months after the first visit were chosen as check points, the rationale of this may need to be better clarified in introduction or in discussion section.

The authors have added a paragraph in the discussion section (lines 201-208) explaining why they differentiate between the short and the long-term, and its implications with the results of the study.

A standard 1 year period was established as a check point for the long-term follow-up. After a preliminary observation of the data, a check point of 4 months after the first visit was established for the short-term, since the authors observed that there was a change of tendency in the amount of visits in the two models approximately 4 months after the first visit.

  1. "resolution" may need to be defined clearly?

The term “resolution” is defined in the Methods section as “the percentage of first visits that did not generate subsequent visits during the next 4 months and during the next 12 months”.

  1. The title of Talbe 1 seemed misising.

Thank you for the remark, the authors have added a title to the Table 1.

  1. What is the difference between tab 3 and tab 4 with totally same title?

Table 3 shows the means of visits generated by each first visit, along with the mean difference and its t-test and the confidence interval at 99%, as well as the and the variation in the means between the face-to-face and the telemedicine model. Table 4 shows the median and the interquartile ranges of the visits generated by each first visit, along with the p-value of the Mann-Whitney test. 

The titles of both Table 3 and 4 are similar but differ in the first word: the first one shows an analysis of the means, whereas the second one shows an analysis of the medians.

  1. The discussion may need to be strengthened, especially the specific manner and their advantages and disadvantages of telemedicine?

The authors have added references to a recently published study stating that telemedicine is beneficial for the care of several chronic diseases. They have also added a discussion on whether other telemedicine solutions may be more effective in reducing follow-up visits than telemedicine visits by phone call.

Reviewer 2 Report (New Reviewer)

Dear authors,

I am having troubles reading this particular version of the manuscript. I am asking for a clean one.

PS: no equations. no plots. no loss function. It is very hard to reach conclusions with none of these important tools.

best regards,

Author Response

Dear authors,

I am having troubles reading this particular version of the manuscript. I am asking for a clean one.

PS: no equations. no plots. no loss function. It is very hard to reach conclusions with none of these important tools.

best regards,

Dear reviewer,

You may have received a version of the manuscript which was edited with the “Track changes” tool, since in a previous round we had already been suggested to make some changes by other reviewers. We don't really understand why in this round the reviewers are "new".

In this version, the only changes made with the “Track changes” tool are the new ones. 

Thanks for your understanding

Round 2

Reviewer 2 Report (New Reviewer)

Dear authors,

I appreciate some of the corrections.

I do not understand the meaning of the p-values in Table 4. What seems to be the problem with the blue/red strike-through words and quantities? I do not get why I have to see them. 0.001 is very different from 0.001.099e-11. Please, try to correct this.

best,

Author Response

Dear reviewer,
In the previous review the scientific notation was changed to the standard p<0.001. The the blue/red strike corresponds to the change control.
Thank you for your attention.

This manuscript is a resubmission of an earlier submission. The following is a list of the peer review reports and author responses from that submission.

Round 1

Reviewer 1 Report

Language Issues

Title:  I recommend “…in a pediatric…” in place of “…in the pediatric…”

“telematic” is not often used in US English.  “Telemedicine visits” is the general term used in the US and in the EC “Market Study on Telemedicine”, which is the author’s first reference in the paper.  (That study also uses the term “teleconsultation,” which the authors might prefer.) The authors also use “telemedicine” in their literature review but switch to “telematic” in line 69.  Is there any difference? 

Lines 84-85, “those for chronic and complex pathologies with monographic agendas (which are expected to require long-term outpatient follow-up).”:  The words “monographic agendas” don’t by themselves communicate either chronic and complex pathologies or an expectation of long-term outpatient follow-up.  While the study’s distinction between general and complex makes great sense, I would rewrite lines 84 and 85 as, “…and those for chronic and complex pathologies which are expected to require long-term outpatient follow-up.”  Then throughout the paper, substitute the use of “monographic agendas” with “chronic and complex pathologies.”  Note, I observed that the tables already do use this suggested nomenclature.  

Reading Notes

Introduction:  The authors do a nice job of summarizing the value of telemedicine in terms of cost-effectiveness.  But their “aim” sentence (lines 71-74) focuses on effect completely dropping cost issues.  That is a major disappointment and reduces the value of the paper. 

Lines 79-80, “nursing visits, dietary visits and those related to diagnostic tests were excluded…”:  I presume this means previously scheduled diagnostic tests.  The authors should clarify that. 

Lines 84-85, “those for chronic and complex pathologies with monographic agendas (which are expected to require long-term outpatient follow-up).”:  The words “monographic agendas” don’t by themselves communicate either chronic and complex pathologies or an expectation of long-term outpatient follow-up.  While the study’s distinction between general and complex makes great sense.  I would rewrite lines 84 and 85 as, “…and those for chronic and complex pathologies which are expected to require long-term outpatient follow-up.”  Then throughout the paper, substitute the use of “monographic agendas” with “chronic and complex pathologies.”  Note, I observed that the tables already do use this suggested nomenclature.  

Lines 88-96:  After several readings, I am still confused by these lines.  To me the degrees of resolution reported in Table 2 are just the proportion of first visits that have no subsequent visits over one of the two specified follow-up visits.  This holds true for both face-to-face and telemedicine visits. 

I would rewrite the paragraph more or less as follows.

The study reports two sets of analyses, one for the degree of resolution and a second for the average numbers of follow-up visits.  The degree of resolution is defined as the percentage of first visits that did not generate subsequent visits.  Eight degrees of resolution were calculated recognizing the differences between telemedicine and face-to-face visits, the 6 and 12 months post-initial visit, and general versus chonic and complex pathologies.  The average number of visits was also calculated for each of the same 8 categories. 

Line 107:  Generally, the 55.85% reported in Table 1 would be rounded up to 56% in any textual reference. 

Substantive Issues

Line 132 to 139:  At least as I have understood the paper, it includes no analyses that would allow this statement.  The previous material suggested that the two time periods were disjoint.  I took that to mean that no patient in the earlier face-to-face time period had a telemedicine option and no patient in the Covid period had a face-to-face option.  While I do recall a mention that telemedicine had been started earlier, I do not recall a statement indicating that face-to-face option was available during the Covid period. 

Of course, it is possible that such overlap existed.  And I do believe the results might be interesting.  But these results cannot be dropped at the end of the paper without at beginning preparing the reader with introductory information and with some simple statistics about the differences in the proportions of mode-switching modes in the two time periods.  If both time periods had both visit options, then the proportions of each need to be communicated and any significant differences need to be reported.

Censoring:  In another data issue, the authors should be clear about the proportion of patient in the Covid period that had not yet completed 12 months since their initial visit.  There’s a sentence that suggests that some censoring occurred because the authors imputed scheduled visits that had not yet occurred.  Indeed, if the data abstraction occurred in May 2021 and the last of the first visits in the Covid period occurred at the end of March 2021, even the 6 month differences are suspect. 

The fact that censoring was superficially glossed over raises my concern that the results might have been biased by the number of Covid-period telemedicine patients who were not followed for anything like 12 months. 

Author Response

Language Issues

Title:  I recommend “…in a pediatric…” in place of “…in the pediatric…”

Dear reviewer, thank you for your suggestions. We have introduced this correction in the text.

“telematic” is not often used in US English.  “Telemedicine visits” is the general term used in the US and in the EC “Market Study on Telemedicine”, which is the author’s first reference in the paper.  (That study also uses the term “teleconsultation,” which the authors might prefer.) The authors also use “telemedicine” in their literature review but switch to “telematic” in line 69.  Is there any difference?

Dear reviewer, thank you for your suggestions. We have introduced these corrections in the text.

Lines 84-85, “those for chronic and complex pathologies with monographic agendas (which are expected to require long-term outpatient follow-up).”:  The words “monographic agendas” don’t by themselves communicate either chronic and complex pathologies or an expectation of long-term outpatient follow-up.  While the study’s distinction between general and complex makes great sense, I would rewrite lines 84 and 85 as, “…and those for chronic and complex pathologies which are expected to require long-term outpatient follow-up.”  Then throughout the paper, substitute the use of “monographic agendas” with “chronic and complex pathologies.”  Note, I observed that the tables already do use this suggested nomenclature. 

Dear reviewer, thank you for your suggestions. We have introduced this correction in the text.

Reading Notes

Introduction:  The authors do a nice job of summarizing the value of telemedicine in terms of cost-effectiveness.  But their “aim” sentence (lines 71-74) focuses on effect completely dropping cost issues.  That is a major disappointment and reduces the value of the paper.

Dear reviewer, thank you for your suggestions. While it is true that the study does not observe the costs of the intervention, as the degree of resolution of the activity is assessed, the use of resources associated with each model is being indirectly taking into account. That is why the title does not talk about costs.

Lines 79-80, “nursing visits, dietary visits and those related to diagnostic tests were excluded…”:  I presume this means previously scheduled diagnostic tests.  The authors should clarify that.

Dear reviewer, thank you for your suggestions. In our study nursing visits, dietary visits and those related to diagnostic tests were excluded, therefore, we only included medical visits. We have explained this point in the text.

Lines 88-96:  After several readings, I am still confused by these lines.  To me the degrees of resolution reported in Table 2 are just the proportion of first visits that have no subsequent visits over one of the two specified follow-up visits.  This holds true for both face-to-face and telemedicine visits. I would rewrite the paragraph more or less as follows.

The study reports two sets of analyses, one for the degree of resolution and a second for the average numbers of follow-up visits.  The degree of resolution is defined as the percentage of first visits that did not generate subsequent visits. Eight degrees of resolution were calculated recognizing the differences between telemedicine and face-to-face visits, the 6 and 12 months post-initial visit, and general versus chronic and complex pathologies.  The average number of visits was also calculated for each of the same 8 categories.

Dear reviewer, thank you for your suggestions. We have introduced this correction in the text.

Line 107:  Generally, the 55.85% reported in Table 1 would be rounded up to 56% in any textual reference.

Dear reviewer, thank you for your suggestions. We have introduced this correction in the text.

Substantive Issues

Line 132 to 139:  At least as I have understood the paper, it includes no analyses that would allow this statement.  The previous material suggested that the two time periods were disjoint.  I took that to mean that no patient in the earlier face-to-face time period had a telemedicine option and no patient in the Covid period had a face-to-face option.  While I do recall a mention that telemedicine had been started earlier, I do not recall a statement indicating that face-to-face option was available during the Covid period.

Of course, it is possible that such overlap existed.  And I do believe the results might be interesting.  But these results cannot be dropped at the end of the paper without at beginning preparing the reader with introductory information and with some simple statistics about the differences in the proportions of mode-switching modes in the two time periods.  If both time periods had both visit options, then the proportions of each need to be communicated and any significant differences need to be reported.

Dear reviewer, thank you for your suggestions. We have introduced a statement (in the last paragraph of the “Methods” section) which indicates that the face-to-face option was available during the Covid period.

Censoring:  In another data issue, the authors should be clear about the proportion of patient in the Covid period that had not yet completed 12 months since their initial visit.  There’s a sentence that suggests that some censoring occurred because the authors imputed scheduled visits that had not yet occurred.  Indeed, if the data abstraction occurred in May 2021 and the last of the first visits in the Covid period occurred at the end of March 2021, even the 6 month differences are suspect.

The fact that censoring was superficially glossed over raises my concern that the results might have been biased by the number of Covid-period telemedicine patients who were not followed for anything like 12 months.

Dear reviewer, thank you for your suggestions. This aspect is part of the study limitations, specifically described in the third paragraph of the discussion.

Reviewer 2 Report

The authors should explain the meaning of ICTs (line number 36). Additionally, the url provided to find more information regarding ICTs is not working (reference number [1])

The degree of resolution is defined as the percentage of first telematics visits that did not generate new visits, however, this metric is used to obtain the degree of resolution of face-to-face model too. The definition should be more general to include both cases of study.

Author Response

Comments and Suggestions for Authors

The authors should explain the meaning of ICTs (line number 36). Additionally, the url provided to find more information regarding ICTs is not working (reference number [1])

Dear reviewer, thank you for your suggestions. The meaning of ICT is described in a specific section at the end of the text (Abbreviations). In relation to the reference, we have verified the link and it works correctly.

The degree of resolution is defined as the percentage of first telematics visits that did not generate new visits, however, this metric is used to obtain the degree of resolution of face-to-face model too. The definition should be more general to include both cases of study.

Dear reviewer, thank you for your suggestions. The concept of "degree of resolution" is the same in the case of the telemedicine model and the face-to-face one. We used identically throughout the text.

Reviewer 3 Report

Dear authors.

The paper is well written, presented and described; as well as the research, discussion and results.

I encourage you to keep working and extending the research with a large sample of patients to support the results shown in this research. 

Author Response

Dear reviewer, thank you for your revision.  

Round 2

Reviewer 1 Report

The authors revisions thoroughly resolved the superficial issues, but failed to provide even the transparency about censoring I requested.  Specifically, they failed to provide a frequency table of the follow-up times in the two study cohorts.